# Improving High-Frequency Details in Cerebellum for Brain MRI Super-Resolution

## Abstract

Deep-learning-based single image super-resolution (SISR) has attracted growing interest in clinical diagnosis, especially in the brain MR imaging field. Conventionally, SISR models are trained using paired low-resolution (LR) and high-resolution (HR) images, and image patches rather than the whole images are fed into the model to prevent hardware memory issues. However, since different brain regions have disparate structures and their size varies, such as the cerebrum and the cerebellum, models trained using image patches could be dominated by the structures of the larger region in the brain and ignore the fine-grained details in smaller areas. In this paper, we first investigate the capacities of several renowned models, by using more blurry LR images than previous studies, as input. Then, we propose a simple yet effective method for the conventional patch-based training strategy by balancing the proportion of patches containing high-frequency details, which makes the model focus more on high-frequency information in tiny regions, especially for the cerebellum. Our method does not depend on model architectures and this paper focuses solely on the T1-weighted brain MR images. Compared with the conventional patch-based training strategy, the resultant super-resolved image from our approach achieves comparable image quality for the whole brain, whereas improves significantly on the high-frequency details in the cerebellum.

## 1 Introduction

High spatial resolution (HR) structural MR images contain detailed anatomical information, which is preferable in clinical diagnosis. However, HR images come at the cost of longer scan time, smaller spatial coverage, and lower signal-to-noise ratio (Plenge et al., 2012), which poses difficulties for the clinical environment due to hardware limits. One solution for this problem is to apply the single image super-resolution (SISR) technique, which only requires a low spatial resolution (LR) image to reconstruct its HR counterpart. It does not require extra scan time or high-cost scanners to generate an HR image and could be used to enhance the image quality from the low-field scanners. Nowadays, SISR has already been applied for medical image diagnosis, such as brain tumor detection (Özyurt et al., 2020).

Training SISR models requires paired LR and HR images so that the models can learn the mapping from the LR to HR images. Due to the rare availability of paired MR images, synthetic LR images, generated from their HR counterpart, have been widely used in the literature. To mimic the behaviors of the real-world low-field scanners, the HR images are typically transformed into the k-space using the Fast Fourier Transform (FFT), and the resolution degradation is performed on the k-space data (Chen et al., 2018b).

In many previous studies, the synthetic LR images are similar to the original HR images, even if only keeping the central 25% k-space data (Chen et al., 2018b;a; Lyu et al., 2020a; Wang et al., 2020a; Li et al., 2021), which may face performance degradation on the real-world LR images. In this paper, we first investigate whether these previously proposed models are capable of learning more complex mappings when using more blurry LR images as input. We follow the procedures precisely in generating the LR images (Chen et al., 2018a;b; Wang et al., 2020a), whereas only keeping 6.25% of the k-space data. By keeping a smaller proportion of the original k-space data, there exists a more significant difference between the synthetic LR and the authentic HR images.

A more fundamental question in brain MRI super-resolution is about the patch-based training strategy. Conventionally, image patches sampled uniformly from the whole brain volume, rather than the whole images, are fed into the model as input to prevent hardware memory issues (Chen et al., 2018a;b; Lyu et al., 2020a; Wang et al., 2020a; Li et al., 2021). However, such uniform sampling introduces a data imbalance issue. The cerebrum, containing low-frequency information, generates far more patches than the cerebellum, whereas the latter contains more complex structures. It could result in the model being more focused on the information from larger regions of the brain and ignoring the fine-grained details in tiny areas. In the literature, the reconstruction of the cerebrum has achieved almost indistinguishable performance using relatively small models (Chen et al., 2018a). However, there is still a huge gap between the super-resolved cerebellum and the authentic HR image, even using decent models (Zhang et al., 2021; Li et al., 2022).

To solve the data-imbalance issue, we propose a simple yet effective method via a non-uniform patch sampling for the conventional patch-based training strategy, which does not depend on model architectures, to treat the whole brain and cerebellum equally to derive better super-resolution quality.

Our main contributions in this paper are:

1. We evaluate the capacity of several renowned models by using more blurry LR images as input, compared with previous studies. By keeping 6.25% of the k-space data rather than 25% used in the literature, the synthetic LR images look more similar to the real-world LR ones, which may improve model generalization ability on real-world images.

2. We propose a direct and effective method using a non-uniform sampling for the patch-based training strategy to improve the reconstruction quality for high-frequency details, especially for the cerebellum. To the best of our knowledge, the proposed method is the first work that treats the brain volume as two separate regions, the cerebellum and non-cerebellum regions. The implementations are available in the GitHub repository.

The paper is organized as follows. Section 2 reviews the development of SISR on natural and MR images. Section 3 introduces the SISR framework and our proposed method for improving the high-frequency details for brain MRI super-resolution. Section 4 explains the settings of the experiments, including dataset and model architectures. Section 5 demonstrates quantitative experiment results and model comparisons. Section 6 summarizes the overall approach to conclude this paper.

## 2 RELATED WORK

### 2.1 DEEP-LEARNING-BASED SUPER-RESOLUTION

Super-resolution (SR) using deep learning models has experienced significant progress over the past decade. Dong *et al.* first proposed SRCNN in 2014, establishing the basic structure for SR models (Dong et al., 2015). Kim *et al.* later proposed VDSR that learns the residual information between the LR and HR images (Kim et al., 2016a), greatly reducing the learning difficulty. Motivated by the success of the ResNet (He et al., 2016), several ResNet-based SR models have been proposed (Zhang et al., 2018b; Ledig et al., 2017), achieving superior performance with the help of the residual connection structure. In addition, the dense connection structure (Huang et al., 2017) proposed by Huang *et al.* has also been applied to SR problems. Tong *et al.* applied several dense blocks and added dense connections between different blocks to build the SRDenseNet (Tong et al., 2017). Zhang *et al.* later took the advantages of both residual and dense connection to propose a residual dense network to further improve the reconstruction quality (Zhang et al., 2018c). With the help of the Transformer models (Vaswani et al., 2017), Lu *et al.* proposed ESRT, which consists of CNN and transformer backbones, achieving competitive results with low computational costs (Lu et al., 2022).

Generative adversarial network (GAN) (Goodfellow et al., 2014) has also been applied to SR problems. Ledig *et al.* proposed SRGAN, which generates more realistic images compared with a CNN-based model (Ledig et al., 2017). Bell-Kligler *et al.* introduced an unsupervised model named KernalGAN (Bell-Kligler et al., 2019), using real LR images for model training. In addition, various model design strategies, such as the learning-based upsampling (Zhang et al., 2018a; Shi et al., 2016), recursive learning (Kim et al., 2016b; Lai et al., 2018) have also been proposed for SR.

However, most of these works are designed for natural 2D images and a direct conversion into their 3D versions generally fails to achieve a good trade-off between the model complexity and accuracy.

## 2.2 SUPER-RESOLUTION ON MR IMAGES

Unlike natural images, structural MR images contain four dimensions, greatly increasing the model complexity as well as learning difficulty. Therefore, models using 2D image slices or 3D image volumes have both been explored in recent years.

2D models are generally more stable and faster to train, whereas they normally fail to learn the information from the third dimension. Zhu *et al.* proposed a 2D multi-scale GAN with a lesion-focused approach to achieve a more stable training process and better perceptual quality for SR images (Zhu et al., 2019). You *et al.* presented a 2D GAN model incorporating cycle consistency and residual learning (You et al., 2019). Du *et al.* built a 2D CNN with residual connections for SR reconstruction of single anisotropic MR images (Du et al., 2020). Lyu *et al.* established an ensemble learning framework using 2D GANs to integrate complementary SR results from each GAN model (Lyu et al., 2020b). Zhao *et al.* introduced a channel splitting block to incorporate different information from different receptive fields to increase the model learning capacity (Zhao et al., 2019).

On the contrary, 3D models can utilize the additional information from the volumetric images and outperform the 2D models by a large margin, at the cost of model complexity. Pham *et al.* first demonstrated that for brain MRI, 3D models outperform 2D models by a large margin (Pham et al., 2017). Chen *et al.* applied the dense connection structure extensively and proposed the 3D mDCSRN model (Chen et al., 2018a), which contains relatively few parameters. A GAN-based model (Chen et al., 2018a) has also been built by Chen *et al.*, producing more perceptually better results. Li *et al.* proposed a lightweight network, called VolumeNet, using separable 2D cross-channel convolutions (Li et al., 2021), which has fewer parameters but still achieves superior performance. Zhang *et al.* adopted the squeeze and excitation architecture (Hu et al., 2018) and attention mechanism to make the model learn from the more informative regions of the brain (Zhang et al., 2021). With the help of the Transformer models (Vaswani et al., 2017), Li *et al.* proposed a multi-scale contextual matching method that can learn information from different scales (Li et al., 2022). Zhou *et al.* applied a domain correction network that can super-resolve unpaired MR images (Zhou et al., 2022).

There is also a bunch of work focusing on different aspects of MRI super-resolution, such as models handling multi-scale LR images (Pham et al., 2019; Dong et al., 2022), image quality transfer (Alexander et al., 2017; Tanno et al., 2021; Lin et al., 2023), and models trained using real-world LR images (Lau et al., 2023; Lin et al., 2023).

Generally, existing works on MRI super-resolution have achieved superior performance. However, even the most advanced models, such as attention-based (Zhang et al., 2021) or transformer-based model (Li et al., 2022), fail to handle the whole brain volume and the cerebellum at the same time. The attention mechanism applied in such models (Zhang et al., 2021; Li et al., 2022) offers an indirect way of learning from more informative areas, whereas the performance is poor on the cerebellum. The non-uniform sampling proposed in this paper provides a naive yet straightforward way to solve this problem efficiently.

## 3 METHOD

### 3.1 BACKGROUND

Training SISR models requires paired LR and HR images. The LR images are typically generated from the corresponding HR images. The resolution downgrading process from the HR image X to the LR image Y could be simplified as:

$$Y = f(X) \tag{1}$$

where $f$ is a certain type of function that causes a loss of image resolution. The SISR problem is to find an inverse mapping function $g$ which could recover the HR image $\hat{X}$ from the LR image Y:

$$\hat{X} = g(Y) = f^{-1}(Y) + r \tag{2}$$

where $r$ is the reconstruction residual.

## 3.2 NON-UNIFORM SAMPLING FOR BRAIN IMAGE SUPER-RESOLUTION

As we previously mentioned in Section 1, the conventional uniform patch sampling results in a data imbalance issue between the over-represented region (cerebrum) and under-represented region (cerebellum), whereas the latter has more complex details and requires much more data to learn.

Ideally, a second model could be trained using only the cerebellum images, which is expected to achieve better results on the cerebellum. However, it requires additional training resources and might introduce problems when merging the two resultant images.

Therefore, a unified approach could be proposed by balancing the number of patches from the cerebrum and cerebellum during the training process. Such non-uniform patch sampling makes the model focus more on under-represented regions. To keep the overall training time unchanged, the number of generated patches stays unchanged, whereas more patches come from the cerebellum compared with conventional patch sampling.

The detailed steps could be summarized as:

1. Extract the cerebellum from the whole brain volume to create a separate cerebellum dataset.
2. Generate patches from the whole brain volume and randomly select 50% of the generated patches.
3. Generate patches from the cerebellum dataset until the number of patches is equal to the whole brain volume patches.
4. Combine the whole brain and cerebellum patches randomly for training.
5. Repeat step 2, 3, and 4 at the beginning of each epoch until the end of training.

The intuition of our approach is straightforward. Since the brainstem contains few patterns, the brain super-resolution focuses more on the cerebrum and cerebellum. Also, as the cerebrum and cerebellum contain disparate structures, it is natural to treat the whole volume super-resolution as two sub-volume super-resolution tasks. Therefore, by keeping a 50/50 data distribution of image patches, the model becomes less biased towards the cerebrum, which naturally helps the reconstruction in the cerebellum. Theoretically speaking, such non-uniform sampling might cause performance degradation in the cerebrum, as less information is used for training. However, in Section 5, we will demonstrate that our modification results in negligible difference compared with conventional uniform sampling.

## 4 EXPERIMENTS

### 4.1 DATASET AND DATA PREPROCESSING

To fairly compare different models and training strategies, we chose a widely used and publicly accessible brain structural MRI database, the Human Connectome Project (HCP) (Van Essen et al., 2013). It contains 3D T1-weighted MR images from 1113 subjects acquired via the Siemens 3T platform using a 32-channel head coil on multiple centers. The images come in high spatial resolution as 0.7 mm isotropic in a matrix size of $320 \times 320 \times 256$. These high-quality images serve as the authentic HR images in the following experiments. We further randomly chose 800 images as the training set, 100 images as the validation set, and the rest 213 images as the test set.

We further performed the Brain Extraction Tool (BET) (Smith, 2002) on the original HR images to delete non-brain tissue to make our model focus only on the brain structures to improve performance.

To generate the corresponding LR images, we followed the same pipeline demonstrated in (Chen et al., 2018a). The detailed steps are as follows:

1. Applying the FFT to authentic HR images to convert the original image into the k-space data.

2. Masking (zeroing) the outer part of the 3D k-space data along two axes representing two MR phase encoding directions.

3. Applying the inverse FFT to the masked k-space data to generate the synthetic LR images.

An illustration of masking the k-space data is shown in Figure 1 on page 5. The above LR image synthesis procedure mimics the real MR image acquisition process where a low-resolution MRI is scanned by reducing acquisition lines in phase and slice encoding directions. It should be noted that the synthetic LR images have the same size as the HR images, avoiding the checkerboard artifacts (Odena et al., 2016).

A hyperparameter, named "scale factor", is introduced in the demonstrated image synthesis pipeline, which determines the proportion of masked k-space data. A larger scale factor masks more k-space data, resulting in more blurry images. In Figure 1 on page 5, we set the scale factor to 2. Therefore, only the central 25% ($1/2 \times 1/2$) of the k-space data is preserved. Using a scale factor of 2 is common in many previous studies (Chen et al., 2018a;b; Li et al., 2021), whereas in this paper, we set the scale factor to 4, the central 6.25% ($1/4 \times 1/4$) of the k-space data, to evaluate the model performance on more blurry LR images. An image comparison using different scale factors is shown in Figure 2 on page 5.

Moreover, as the proposed non-uniform sampling, described in Section 3.2, requires a separate sampling on the cerebellum, a separate dataset containing the cerebellum is created. It is achieved by using FastSurfer (Henschel et al., 2020), a deep-learning-based open-sourced brain segmentation software [1]. It should be stressed that segmentation masks are generated using only the authentic HR images, and the corresponding masks will be applied on both HR and LR images to ensure both HR and LR cerebellums are also voxelwise paired. An illustration of the cerebellum extraction pipeline is shown in Figure 3 on page 6.

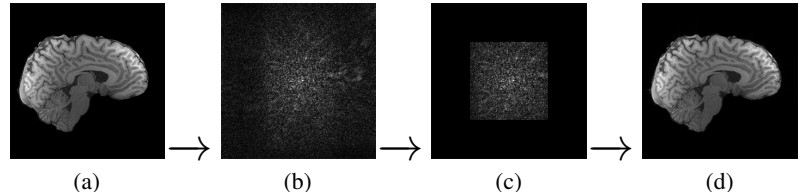

|  (a)  |  (b)  |  (c)  |  (d)  |

Figure 1: An illustration of masking the k-space data. (a) represents the HR image. (b) is obtained by applying FFT to (a). (c) is obtained by masking (zeroing) the phase and slice encoding directions of (b) and (d) is generated by applying inverse FFT on (c). We set the scale factor to 2.

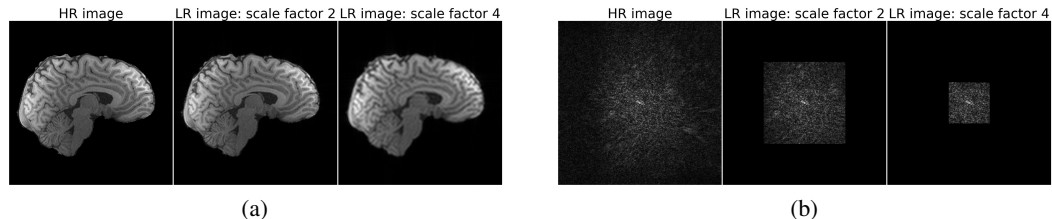

|  (a)  |  (b)  |

Figure 2: MR image and corresponding k-space data comparisons using different scale factors. In (a), from left to right, it shows the original HR image, LR image using a scale factor of 2 and 4. (b) shows the corresponding k-space data using different scale factors. For scale factors 2 and 4, the central 25% and 6.25% of the data are preserved.

## 4.2 MODELS

As the proposed method focuses primarily on the patch sampling strategy, there is no limitation on the choice of model architectures. Therefore, we chose two widely recognized models in this

---

[1]https://github.com/Deep-MI/FastSurfer

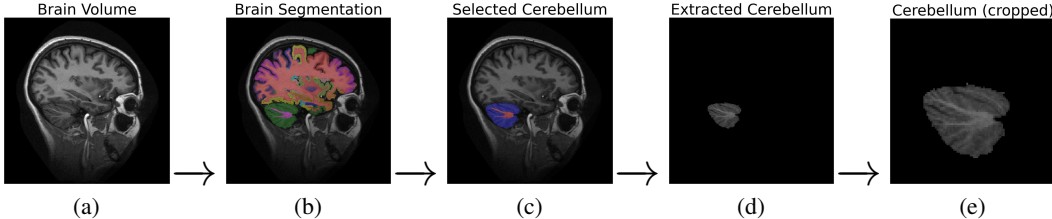

| Brain Volume | Brain Segmentation | Selected Cerebellum | Extracted Cerebellum | Cerebellum (cropped) |
| :---: | :---: | :---: | :---: | :---: |
| (a) | (b) | (c) | (d) | (e) |

Figure 3: An illustration of the cerebellum extraction pipeline using FastSurfer. (a) represents the authentic HR image. (b) is obtained by applying FastSurfer to (a). (c) is obtained by selecting the cerebellum-related mask of (b) and (d) is generated by applying the cerebellum mask (c) on (a). Finally, we remove useless background in (d) to derive (e).

field, mDCSRN (Chen et al., 2018b) and mDCSRN-WGAN (Chen et al., 2018a) to evaluate the proposed approach. The main reason behind the above architecture choice is that they both choose the DenseNet (Huang et al., 2017) as the backbone, which is still popular in various model designs. Also, they represent two distinct model designs, CNN-based and GAN-based models.

The mDCSRN model adopts a DenseNet (Huang et al., 2017) architecture with 4 dense blocks. The mDCSRN-WGAN model uses the same architecture of mDCSRN as the generator and uses Wasserstein GAN structure (Arjovsky et al., 2017) to guide the model training.

### 4.3 TRAINING AND TESTING

In the training process, Adam optimizer (Kingma & Ba, 2014) is used as the default optimizer for all models. The initial learning rate is set to 0.001 and then multiplied by 0.5 every 10 epochs. The batch size is set to 64 as default and the total number of epochs is set to 50. We use the L1 loss for mDCSRN and the generator of the mDCSRN-WGAN model since L1 loss has been shown to generate more realistic images than L2 loss (Zhao et al., 2016). The size of image patches is set to $32 \times 32 \times 32$ for all patches.

To evaluate the super-resolution performance, we used the structural similarity index (SSIM), peak signal-to-noise ratio (PSNR), and normalized root mean squared error (NRMSE) to measure the similarity between super-resolved (SR) images and HR images. SSIM uses the mean, variance, and covariance to estimate the similarity of the two images. PSNR is used to further quantify the recovered image quality using the mean squared loss. NRMSE is a more direct way to measure the pixel-wise similarity between the original and super-resolved images. In general, lower NRMAE, higher PSNR, and higher SSIM values represent better super-resolution results. Although these metrics have been criticized that a high score does not represent a better image quality (Wang et al., 2020b), they are still the most commonly used metrics, and no other perceptual quality metrics for 3D MR images have been applied in the literature.

## 5 RESULTS

In this section, we first demonstrate the SISR model performance using synthetic LR images, generated under a scale factor of 4. Then, we compare the model performance between the conventional patch sampling and the proposed non-uniform sampling. Due to space constraints, we only include sample output images from the mDCSRN model as a demonstration.

### 5.1 MODEL PERFORMANCE UNDER A SCALE FACTOR OF 4

Figure 4 on page 7 demonstrates the model performance of the whole brain volume using the mDCSRN model from three perspectives. It could be observed that the SR image is similar to the HR image in most regions of the brain, especially in the cerebrum. It indicates that relatively simple models, such as mDCSRN, are still capable of handling the more blurry patterns, especially in the cerebrum region. The quantitative analysis is demonstrated in Table 1 on page 7. From Table 1 on

page 7, it could be observed that both mDCSRN and mDCSRN-WGAN models significantly improve the image quality in all three metrics.

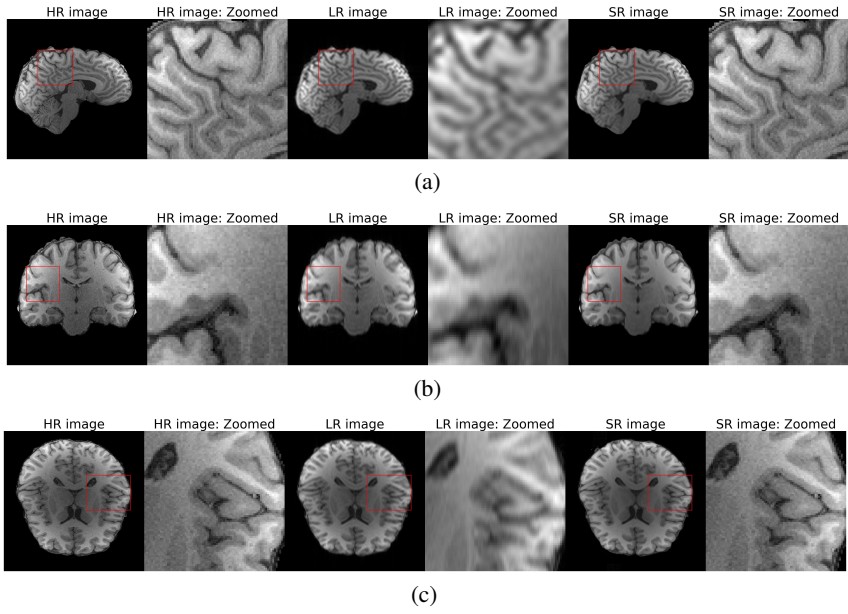

(a)

(b)

(c)

Figure 4: An illustration of the HR, LR, and SR brain image using the mDCSRN model. The zoomed images show the region of interest from the whole brain volume using a red bounding box. (a) shows the sagittal view, (b) shows the coronal view, and (c) shows the axial view.

Table 1: Model performance of the whole brain

| Model | Image Pair | PSNR | SSIM | NRMSE |
|---|---|---|---|---|
| None (Baseline) | HR & LR | 32.33 ± 2.92 | 0.92 ± 0.01 | 0.148 ± 0.09 |
| mDCSRN | HR & SR | **34.23 ± 2.14** | **0.95 ± 0.01** | **0.119 ± 0.06** |
| mDCSRN -WGAN | HR & SR | **34.19 ± 2.11** | **0.95 ± 0.02** | **0.121 ± 0.07** |

## 5.2 COMPARISONS BETWEEN CONVENTIONAL SAMPLING AND THE PROPOSED APPROACH

Table 2 on page 8 and Table 3 on page 8 demonstrate the quantitative analysis of model performance for mDCSRN and mDCSRN-WGAN respectively. For each model, we compare the original sampling approach and the proposed approach on the whole brain volume and the cerebellum respectively.

From Table 2 on page 8 and Table 3 on page 8, it could be found that when considering the reconstruction quality of the whole brain volume for both the mDCSRN and mDCSRN-WGAN model, the conventional patch-based training approach achieves similar performance with our sampling approach in all three metrics. Whereas our approach significantly surpasses the conventional patch-based approach, when we only focus on the reconstruction quality of the cerebellum for both models.

The reason for the improved performance on the cerebellum is clear, as we have increased the proportion of image patches from the cerebellum for models to learn. The potential reason for the comparable performance on the whole brain volume between the conventional and the proposed approach is that the number of patches from the whole brain is redundant. Reducing the proportion of patches from the over-represented regions does not affect model performance significantly. Also, at the beginning of each epoch, random patches will be selected until the end of training. Therefore, the model is still able to learn information from all whole brain patches.

Figure 5 on page 8 and Figure 6 on page 9 demonstrate the super-resolution performance between the conventional approach and the proposed approach on both whole brain volume and cerebellum

respectively. It could be observed that our approach achieves almost the same performance on the whole brain volume, whereas reconstructs more details on the cerebellum.

Therefore, compared with the conventional sampling approach in the literature, the proposed non-uniform sampling approach achieves comparable performance on the whole brain volume, whereas significantly improves the reconstruction quality for the fine structures in the cerebellum.

Table 2: mDCSRN model performance of the whole brain and cerebellum

| Brain Regions | Image Pair | PSNR | SSIM | NRMSE |
|---|---|---|---|---|
| Whole Brain | HR & LR (baseline) | 32.33 ± 2.92 | 0.92 ± 0.01 | 0.148 ± 0.09 |
| | HR & SR (conventional sampling approach) | **34.23 ± 2.14** | **0.95 ± 0.01** | **0.119 ± 0.06** |
| | HR & SR (our approach) | **34.18 ± 1.98** | **0.95 ± 0.02** | **0.120 ± 0.05** |
| Cerebellum | HR & LR (baseline) | 28.37 ± 2.66 | 0.88 ± 0.01 | 0.15 ± 0.10 |
| | HR & SR (conventional sampling approach) | 28.89 ± 1.81 | 0.905 ± 0.01 | 0.135 ± 0.06 |
| | HR & SR (our approach) | **29.62 ± 1.55** | **0.93 ± 0.01** | **0.127 ± 0.02** |

Table 3: mDCSRN-WGAN model performance of the whole brain and cerebellum

| Brain Regions | Image Pair | PSNR | SSIM | NRMSE |
|---|---|---|---|---|
| Whole Brain | HR & LR (baseline) | 32.33 ± 2.92 | 0.92 ± 0.01 | 0.148 ± 0.09 |
| | HR & SR (conventional sampling approach) | **34.19 ± 2.11** | **0.95 ± 0.02** | **0.121 ± 0.07** |
| | HR & SR (our approach) | **34.13 ± 2.17** | **0.95 ± 0.03** | **0.122 ± 0.05** |
| Cerebellum | HR & LR (baseline) | 28.37 ± 2.66 | 0.88 ± 0.01 | 0.15 ± 0.10 |
| | HR & SR (conventional sampling approach) | 28.91 ± 1.73 | 0.902 ± 0.02 | 0.139 ± 0.04 |
| | HR & SR (our approach) | **29.47 ± 1.81** | **0.918 ± 0.02** | **0.124 ± 0.05** |

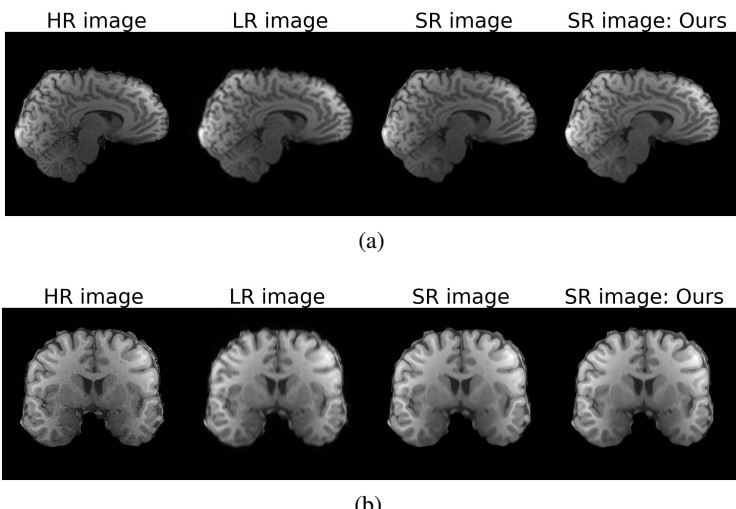

Figure 5: An illustration of the HR, LR, SR (using conventional sampling), and SR (using the proposed sampling) whole brain image by mDCSRN model. (a) shows the sagittal view and (b) shows the coronal view.

## 6 DISCUSSION AND CONCLUSION

We have proposed a simple yet effective method, via a non-uniform sampling strategy, for the conventional patch-based training in brain MRI super-resolution problems. We provide a different perspective by observing the brain structures in super-resolving brain MR images. By maintaining a 50/50 data distribution of patches, our approach achieves comparable reconstruction performance on

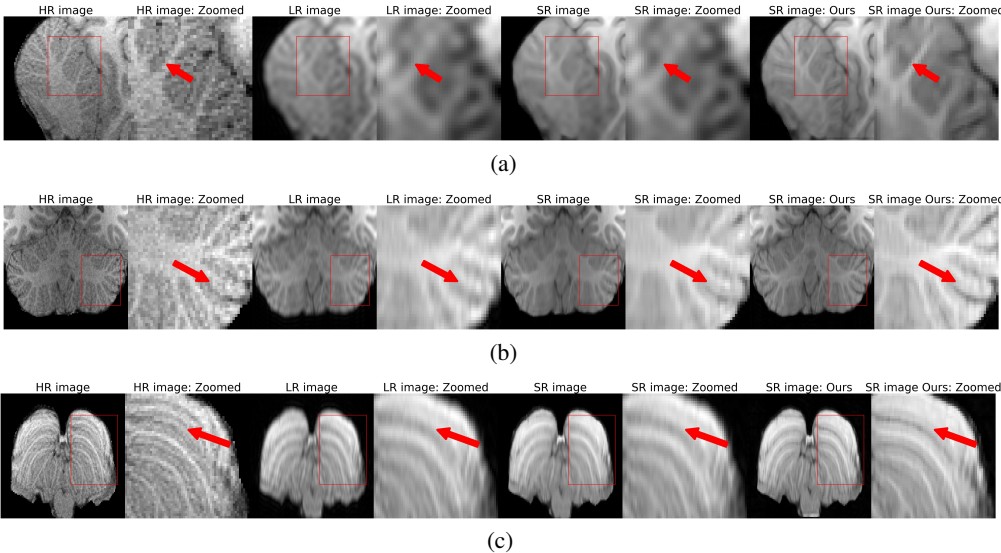

Figure 6: An illustration of the HR, LR, SR (using conventional sampling), and SR (using the proposed sampling) cerebellum image by mDCSRN model. The zoomed images show the region of interest from the cerebellum using a red bounding box. (a) shows the sagittal view, (b) shows the coronal view, and (c) shows the axial view.

the whole brain volume, whereas significantly surpasses the conventional approach on the cerebellum. More high-frequency details in the cerebellum have been restored from our approach.

We have demonstrated the capacity of several renowned models by using much more blurry LR images as input in Section 5. In the literature, a scale factor of 2 has been widely adopted to generate LR images. In this paper, we select a scale factor of 4 to generate LR images and the models are still capable of learning the mappings between the LR and HR images.

We have shown that our non-uniform sampling strategy across the whole brain volume does not result in significant reconstruction degradation in the non-cerebellum regions. The most probable reason behind this is that in the conventional approach, the number of patches from the over-represented regions is already redundant, and non-cerebellum regions contain simpler patterns. As we demonstrated in Figure 4 on page 7, Table 2 on page 8 and Table 3 on page 8, the mDCSRN model can achieve indistinguishable reconstruction quality over non-cerebellum regions.

There are still some aspects that require further improvements. Firstly, although our approach focuses on data balancing and does not depend on model architectures, it might be worthwhile to apply such sampling to more advanced architectures. Also, our proposed sampling strategy is mainly based on intuitions. Given that the brain mainly contains the cerebrum, cerebellum, and brainstem, whereas the latter contains very few structures, it is reasonable to treat the whole brain volume SR into two sub-volume SR problems and apply a 50/50 data distribution for the cerebellum and non-cerebellum regions. However, more sophisticated sampling methods could be explored to evaluate the optimal combination of the two regions.

To conclude, we propose a more effective sampling method for the conventional patch-based training strategy for brain MRI super-resolution problems. By using a non-uniform sampling across the whole brain volume, the model tends to learn more high-frequency details from the cerebellum, rather than being focused on non-cerebellum regions due to data imbalance. With the help of the proposed sampling approach, we improve the model performance on the cerebellum significantly and also achieve comparable performance on the whole brain volume.

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
