# OpenReview forum: "Improving High-Frequency Details in Cerebellum for Brain MRI Super-Resolution"
_ICLR.cc/2024/Conference — Submitted to ICLR 2024_

### Official Review · Reviewer_rbC8 · 2023-10-31

**Soundness:** 2 fair
**Presentation:** 2 fair
**Contribution:** 1 poor
**Rating:** 3
**Confidence:** 5

**Summary:**

The presented paper concerns the task of MRI super-resolution. The specific task is to improve details in the smaller brain regions of the cerebellum. The methodological contribution is an alternative to conventional patching in training, where the authors first separated the cerebellum from the entire brain. Next, they generate patches from the complete brain volume, selecting half of them at random. Concurrently, patches were produced from the cerebellum dataset until their count equaled that of the whole brain volume patches. Finally, patches from both sources were combined in a random manner. These steps are repeated at each epoch.

In experiments, the authors show improved results on the cerebellum.

**Strengths:**

- well motivated method for medical image analysis application
- the paper is easy to follow

**Weaknesses:**

The clear weakness is the lack of a technical, methodological, or experimental contribution that aligns with the standards of acceptance for ICLR. For example, the patching strategy is a trivial balancing of underrepresented classes in training. Overall, I think the idea can be a useful trick in the bag for this specific application and therefore could be relevant to the applied medical image analysis community.

**Questions:**

-

---

### Official Review · Reviewer_V9Gd · 2023-10-31

**Soundness:** 1 poor
**Presentation:** 3 good
**Contribution:** 3 good
**Rating:** 1
**Confidence:** 2

**Summary:**

The work aims to investigate effects of two changes to the dataset: 1. Lowering the resolution of the input from a convention of 25% to 6.25% and 2. Reducing the portion of the patches created from the cerebrum of the brain to introduce balance between the cerebrum and cerebellum when training patch based Super Resolution models. Results indicate that the CNN and GAN based models used can still work with a lower resolution image it is typically trained on, and balancing the patch portions can increase the accuracy in the cerebellum with small effect on the cerebrum.

**Strengths:**

Overall analysis is accurate and the questions asked are important factors that should be considered by the community when training such models.

**Weaknesses:**

First of all, even though the paper mentions a github repository of their model, we have not received the mentioned code in any way.

The paper tries to answer two questions, which are both incompletely answered. While they claim that using 6.25% of the k-space is closer to real life low resolution images, they do not provide any evidence to prove their claim. They also aim to argue the generalizability of the model on a lower resolution than previous works. However, for this to be investigated, various scales of k-space should be considered and experimented on. The effect of removing a portion of the k-space should also be discussed, since the lower the resolution, the model has to synthesize more information, which could be critical when dealing with detailed, or subject varying structures.

Regarding solving the imbalance of patches, if patches are created after extracting the cerebellum through segmentation, there will be a complete loss of patches that has an overlap between the cerebrum and the cerebellum. There should be some explanation or discussion on this issue. Also, even though they did mention that there should be more investigation than using just a 50-50 balancing scheme, such work should be included in the main work. Since this is an analysis on how to create the patches, an analysis/investigation on the size of patches should be included as well. Since we have the previous question, how the resolution of the inputs affect this experiment should be looked into.

Finally, even though the introduction and related work section clearly mentions transformer based models, there is no analysis on such models or any explanation of why only CNN and GAN based models were used for analysis.

Overall, while both topics investigated by the authors are important, both analysis are incomplete and lack reasoning behind several factors used/effected in the experiments.

**Questions:**

Reasoning behind the experiment setup could be clarified. The limitations of the study, as mentioned in weakness should be discussed. Both topics need more experiments than investigating a single training session.

---

### Official Review · Reviewer_Mibw · 2023-11-01

**Soundness:** 1 poor
**Presentation:** 1 poor
**Contribution:** 1 poor
**Rating:** 1
**Confidence:** 5

**Summary:**

The Authors propose a non-uniform patch sampling for the super-resolution of T1-weighted magnetic resonance imaging data.

The paper does not present the contributions and quality to be accepted to the ICLR conference. I suggest the Authors submit a short ISMRM abstract from this work.

**Strengths:**

The motivation behind the paper has been defined correctly, i.e., the super-resolution (SR) algorithms might affect fine-grained parts of the brain, such as the cerebellum. The Authors propose a non-uniform patch sampling for the CNN patch-based SR.

The state of the art has been written correctly.

**Weaknesses:**

The method has not been validated lege artis, i.e., the Authors assume the data has been sampled via the Cartesian sampling. Also, they neglect the phase of the signal once transforming the data to the k-space.

Btw. – KernelGAN, not "KernalGAN"

**Questions:**

"Masking (zeroing) the outer part of the 3D k-space data along two axes representing two MR phase encoding directions." -- Why do the Authors assume the reconstruction has been done via the standard Cartesian sampling of the k-space?

What if the acquisition has been performed via accelerated methods like SENSE or GRAPPA? Indeed, the HCP data has been acquired with the parallel method.

---

### Official Review · Reviewer_Pn3Y · 2023-11-01

**Soundness:** 1 poor
**Presentation:** 1 poor
**Contribution:** 1 poor
**Rating:** 3
**Confidence:** 4

**Summary:**

Summary:
This paper proposes a balanced sampling strategy to emphasize cerebellum structure in brain MRI super-resolution.

**Strengths:**

Strength:
Qualitative results in Fig 6 show improved structure in the cerebellum region.

**Weaknesses:**

Major Comments:

- The technical contribution of this paper is not significant since the author just augmented the dataset with a specific part of the brain images. It is well known in the machine learning community that increasing under-represented samples results in improved performance for those categories. This shortcoming can not be fixed within the review period.
- The authors took out-of-the-box super-resolution architecture and only experimented with the sample augmentation strategy, which does not suffice for ICLR publication.
- My suggestion to the authors is to polish the manuscript by clearly emphasizing the importance of cerebellum super-resolution and showing its use case in clinical application. It could be an interesting publication in a medical venue such as Medical Image Analysis, MELBA, or similar other journals.
- It is not clear why row one of Table 1 has HR & LR and the other rows have HR & SR. Also, what is the Baseline model? Zero filling in k-space?
- Figure 4. seems to have a major bug. The HR and SR image columns look identical. Authors should check that with care.

**Questions:**

see weakness

---

### Meta-Review · Area_Chair_4Dtp · 2023-12-12

**Metareview:**

Qualitative results in Fig 6 show some improvement.

The technical contribution of this paper is not significant since the authors just augmented the dataset with a specific part of the brain images. Some parts of the tables and figures need to be further clarified. The method has not been validated lege artis, i.e., the Authors assume the data has been sampled via the Cartesian sampling. Reasoning behind the experiment setup could be clarified

All reviewers provide negative scores. The authors did not respond.

**Justification For Why Not Higher Score:**

N/A

**Justification For Why Not Lower Score:**

N/A

---

### Decision · Program_Chairs · 2024-01-16

Reject